# Hemp: From Field to Fiber—A Review

**João Mariz, Catarina Guise, Teresa Luísa Silva** ⓘ **, Lúcia Rodrigues \* and Carla Joana Silva** ⓘ

CITEVE—Technological Centre for Textile and Clothing of Portugal, Rua Fernando Mesquita, 2785, 4760-034 Vila Nova de Famalicão, Portugal; jmariz@citeve.pt (J.M.); cguise@citeve.pt (C.G.); tsilva@citeve.pt (T.L.S.); cjsilva@citeve.pt (C.J.S.)

\* Correspondence: lrodrigues@citeve.pt

**Abstract:** Hemp fibers derived from *Cannabis sativa* L. have experienced a resurgence in popularity over the past few decades, establishing themselves as one of the most sought-after fibers. This article delves into the intricacies of the hemp production chain, offering a comprehensive understanding from field to fiber. Key aspects covered include the botany of hemp, cultivation requirements, the impact of various factors on plant growth, the harvesting process, different methods of fiber extraction, fibers properties, and suitable spinning processes. Recent studies of hemp's Life Cycle Assessment are explored, shedding light on how it compares to other sustainable crops and providing insights into the true sustainability of hemp, substantiated by numerical data. The article also addresses challenges encountered throughout the hemp production chain and speculates on future directions that may unfold in the coming years. The overall goal of this study is to provide a knowledge base encompassing every facet of hemp fiber production. It elucidates how different technological approaches and the technical properties of fibers play pivotal roles in determining their ultimate applications. By offering a comprehensive overview, this article contributes to the broader understanding of hemp as a valuable and sustainable resource in the textile industry.

**Keywords:** *Cannabis sativa* L.; hemp fibers; Life Cycle Assessment; sustainability; textile industry

## 1. Introduction

The textile industry stands as one of the largest global industrial sectors, having a market value of around 1 trillion USD [1]. However, it is also recognized as one of the most polluting industries, contributing to issues such as water pollution, greenhouse gas emissions, and land occupation [2]. The emergence of Fast Fashion at the end of the 20th century, characterized by the low-cost production of new clothes, has significantly increased clothing production, leading to a surge in waste generated by both textile companies and consumers [2,3].

In quantifiable terms, the textile industry is estimated to annually consume 79 billion $m^3$ of water. For instance, the production of a single cotton t-shirt requires around 2700 L, equivalent to the necessary drinking water for an individual for 2 and ½ years [2,4]. Moreover, a substantial concern revolves around the release of microplastics, as the washing of clothing made from synthetic fibers contributes to the annual release of over half a ton of microplastics, constituting approximately 35% of all primary microplastics in ecosystems [2,3]. Additionally, the substantial generation of textile residues poses a significant environmental risk, with approximately around 87% of all textile products ending up in landfills or being incinerated as of 2022. This translates to an annual production of 11 kg of thrash per individual in the European Union (EU) [2].

Nowadays, the fashion and textile market is dominated by the use of polyester (around 54%) and cotton (with a market share of 22%) [1]. Conventional synthetic fibers, including polyester, are produced using substantial amounts of nonrenewable resources and are not biodegradable [5,6]. Similarly, the production of cotton, despite being a natural fiber,

involves the intensive use of water, fertilizers, pesticides, and other chemicals, causing significant environmental damage to the soil and to groundwater reservoirs [5–7].

Considering the imperative to reduce carbon emissions, focusing on environmental sustainability and develop strategies for durable, reusable, and recyclable textiles, there has been a renewed interest in the research and development of new alternative natural fibers [2,8]. Among these alternatives, industrial hemp (*Cannabis sativa* L.) emerges as a key fiber in supporting the textile industry's journey towards sustainability. Currently, hemp has a market share of textile fibers of 0.3% [1,9].

As a result, this article concentrates on reviewing the process of obtaining hemp fibers and explores opportunities reported in the literature that could enhance and standardize the process of obtaining hemp fibers.

## 2. Discussion

### 2.1. Industrial Hemp

The cultivation of hemp (*Cannabis sativa* L.) (Figure 1a) is one of the oldest domesticated plantations in the world, dating back to China around 2700 BC, and is mainly used to produce paper, ropes, food, medicines, cosmetics, and textiles (Figure 1b) [9]. This wide range of applications makes hemp a unique plant that can have positive impacts in many industries [10]. The production of hemp reached its peak in the mid-20th century; however, the demand has decreased considerably due to the regulation and the development of cheaper materials. Moreover, the cultivation of hemp was prohibited in several countries due to the association with the production of illegal substances [11–13]. In this context, it is important to separate industrial hemp (fiber and food) and the narcotic variation of the Cannabis genus [11]. Therefore, biologically, hemp makes up part of a family (*Cannabaceae*) and genus (*Cannabis*) of plants with a wide assortment of chemicals [12]. Therefore, the only parameter that is used to separate industrial hemp from the medicinal one is the level of the psychoactive cannabinoid tetrahydrocannabinol (THC) [11]. Thus, industrial hemp is defined, in most countries and in the EU, as a variety of the species *Cannabis sativa* L., that has a percentage level under 0.3% of THC [12,14].

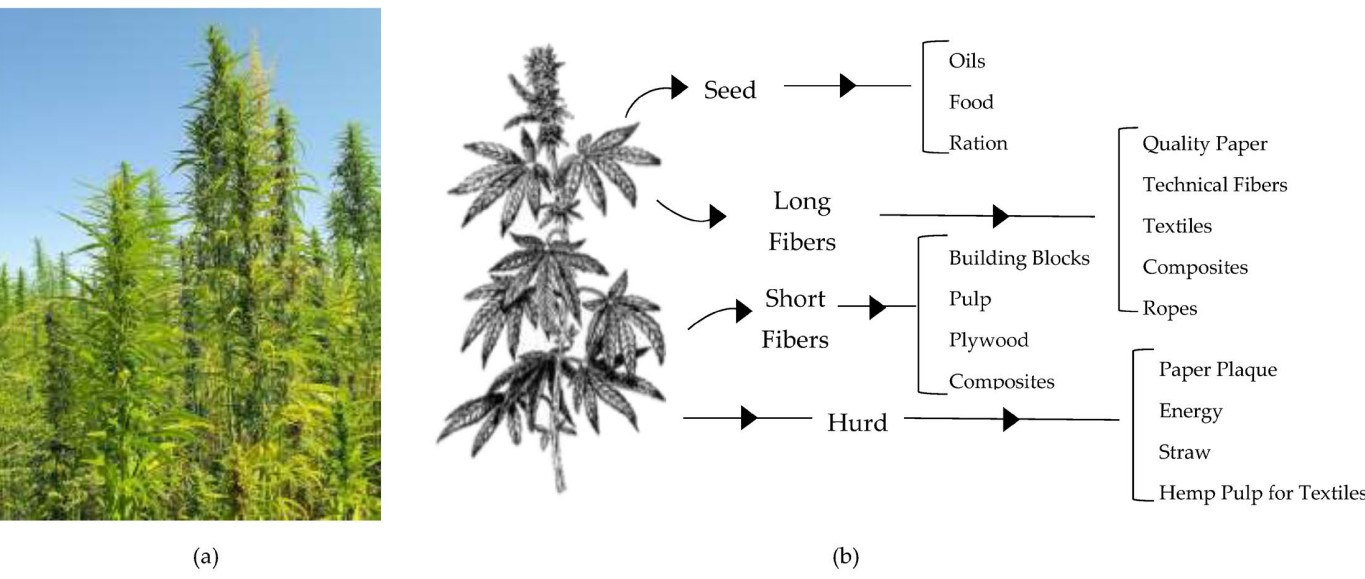

(a) (b)

**Figure 1.** (**a**) Plantation of *Cannabis sativa* L. (**b**) Some of the different applications of *Cannabis sativa* L, divided by the source of the materials. Based on [9].

Hemp is habitually a dioecious plant, which obligates a cross-pollination with a diploid genome; however, monoecious varieties have been explored [11]. This leads to female and male plants being valued at different levels. For medicinal purposes, an all-around plantation of female plants is more desirable, whereas, for fiber production, males are

preferred. In seed production, a surplus of female plants is desired, with few male plants. As referred before, monoecious varieties have been tapped into, as this could achieve a higher yield in production [15].

At the physiological level, the hemp stalk could reach between heights of 1.5 m and 5 m and have a diameter of 5 to 15 mm; this leads to a yield of fibers around 25% to 40% of the total weight of the plant [12,16]. Therefore, ~75% of the stalk is hurds, having many applications, such as the ones referred to in Figure 2 but also hemp pulp for textile applications [17,18]. Additionally, and in physiological terms, hemp roots have a well-developed system, reaching depths of 1 m and producing natural and organic channels, allowing the access of air, water, and the release of gases [15]. Thus, this network of roots allows the use of hemp as an agent for the phytoremediation of damaged soils, also making hemp an excellent candidate as a rotation plant by farmers [12,15]. The latter is already in use, where, in China, farmers use hemp as a rotation plant with crops like soybeans, tobacco, wheat, and corn [12]. Furthermore, hemp is an annual plant, which, along with its root system, has a sophisticated complex of leaves and is one of the fastest-growing plants in existence [19]. This fact permits hemp to absorb close to 10 tons of $CO_2$ per hectare from the atmosphere during a single cycle of production, thus improving air quality and having a positive impact on the environment [19]. Consequently, hemp could be seen as a crop that could help achieve the goals set by the EU Climate Pact and European textile strategies to help fight climate change, such as the reduction of greenhouse gas emissions, promotion of a circular economy, natural resource management, and substitution of fossil fuel products [19,20]. In summary, the upcoming section is poised to provide a comprehensive and detailed examination of the various facets of hemp production. It will likely contribute to an understanding of the current state of the hemp industry, its potential for growth, and the hurdles that need to be overcome for sustainable and successful hemp cultivation.

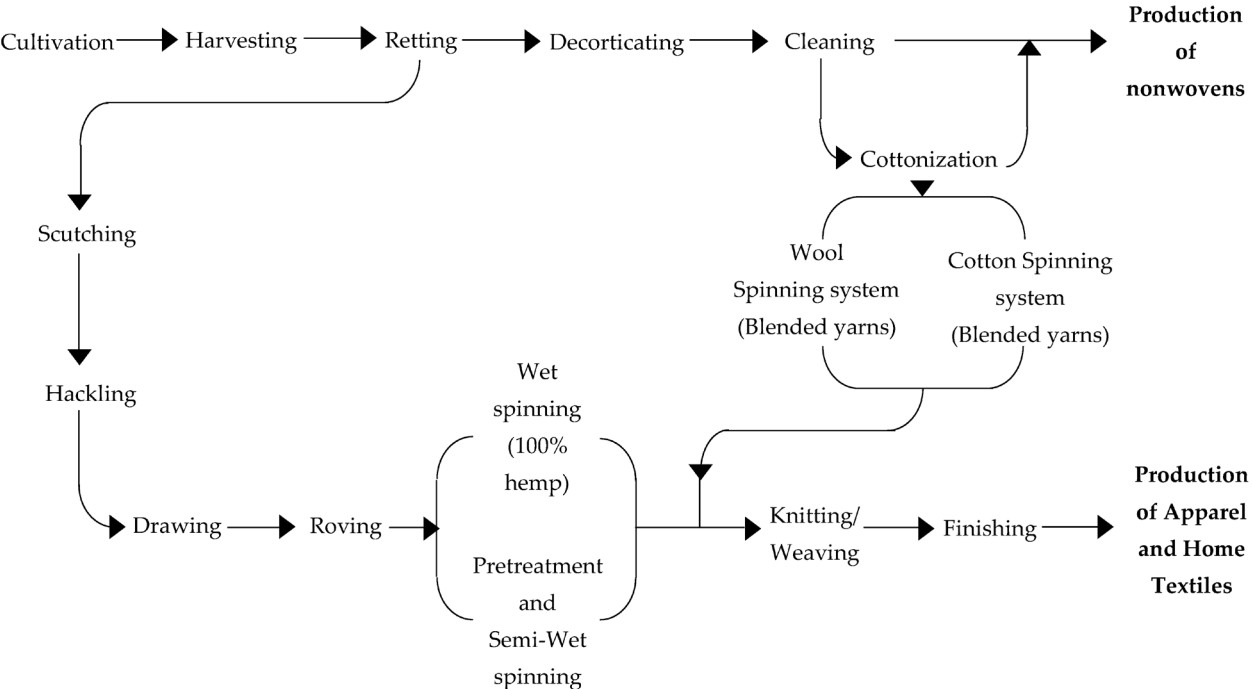

**Figure 2.** Different stages of hemp processing from the field to the final product. Based on [12,19].

### 2.2. Growing Industrial Hemp

The production of hemp as a whole is well reported in the United Nations Food and Agriculture Organization (FAO), which gives free information about food and agriculture statistics in every country in the world [21]. Although, according to [12], the information

available in the database is incomplete and does not include countries that produce hemp oriented toward textile fibers, like the United States of America and Canada, as well as other major players in this field of hemp production. Therefore, what it is possible to interpolate from the data obtained by the FAOSTAT (FAO's Corporate Statistical Database) in 2021 (most recent data available) is that the global estimate of hemp production was 287,318 tons. This represents a total area of 74,307 hectares in 20 different countries, where the five top producers of hemp, as of 2021, were France (143,110 tons), China (72,878 tons), North Korea (15,096 tons), Poland (15,080 tons), and the Netherlands (13,280 tons) [20]. While looking through the data, it is important to notice the lack of mention of hemp production in the United States of America, even with literature and news about its production and development in the country in question. Hence, according to the USDA (United States Department of Agriculture), in 2021, there was a total production of hemp, in all fields of application, of 26,397 tons, which would make the United States the third-largest producer in the world [22]. This anomaly in the data presented by the FAO institution is strange due to the absence of reference to the USA, although this fact is also reported in the literature [12]. Furthermore, in consonance with the EU, in 2022, the total volume of production of hemp in the EU was 179,020 tons, with France responsible for more than 60% of all production, followed by Germany with 17% and the Netherlands with 5% [23]. However, this information is not mentioned in FAOSTAT, where Germany does not even show up as a top 10 producer of hemp. The accuracy and completeness of hemp production data urge a more thorough investigation into the discrepancies observed in reporting between different agricultural organizations and databases.

Since the 1960s, global hemp production has experienced a noteworthy trend. The lowest point was observed in 1990, when the annual production reached 83,997 tons. However, there has been a consistent and visible upswing in global hemp production since then, and remarkably, this increase occurred without a proportional expansion of the harvested area [12,21,23]. Despite a 35% decrease in the harvested area since 1990, there has been an impressive 242% surge in the overall global production of hemp [21]. This means a significant increase in the yield, which has grown rapidly in the last few years—more specifically, since 2015 [21]. Moreover, in 2017, the average yield of hemp fiber extracted per hectare was 4012 kg/ha, while, in 2021, individual countries such as Italy, France, and the Netherlands all had yields of over 7850 kg/ha [12]. Additionally, the resurgence of knowledge that had been lost during the legal restrictions imposed over many decades has played a crucial role in this positive trend [24].

In terms of agriculture, hemp grows best in sandy loam-type soils with good water retention, a balanced soil in nutrients, and at temperatures between 16 and 27 °C [11]. This means that, to obtain high-quality fiber, good land and sufficient water must be used, contrary to the idea that, for hemp growth, minimal input is necessary when a high-quality fiber is wanted [24]. Although, it has been reported that the need for water and fertilization is lower than for other fiber-oriented plants such as cotton or flax, as less than half of the water is needed for the plants previously referred. In full detail, is estimated that hemp needs between 250 and 700 mm of water over the entire growing season, which is critical during establishment, whereas, after this stage, the plant can survive with less water (250–3350 mm) due to its root system [25]. Also, due to its fast growth, it is a natural weed killer and inhibits or lowers the levels of fungi and nematodes in the soil, thus being able to flourish without pesticides or fungicides [10,11,26]. Additionally, hemp is a plant with a short day cycle, and its light period is critical, which means that the cultivation of hemp is limited by latitude, as, in Europe, latitudes between 20° and 40° are considered ideal for hemp growth [25,27]. Due to this fact, hemp seed in Europe is generally sowed in the spring in mid-March to late May, although, this can vary according to the weather and location. The harvesting timing depends highly on the purpose of the plantation, where fiber-oriented hemp is harvested just before the flowering, so 3 months after sowing, around late July and, for oils or other applications like CBD, in October [24]. Furthermore, at the farm level, hemp plantation is similar to other row crops, where it can be drilled or planted in very tight

rows [17]. In addition, hemp can grow in very high-density seed rates, which, in the case of fiber-oriented plantations, is desired, because it leads to higher stalk growth between 5 and 6 m tall, depending on the variety [11]. If the focus is to obtain oilseed and CBD, the density should be lower and further apart to promote the growth of branches and flowers [11]. The seed drillers used for hemp usually drill at depths of 2 to 3 cm and a distance between lines of 9 to 17 cm. The density of the driller varies considerably, from 40 to 150 kg/ha. According to the literature, densities of 40 to 65 kg/ha are recommended for hemp, reaching around 200 to 300 plants per square meter. However, for hemp focused on seed production, this density should be lowered to around 20 kg/ha [27]. Additionally, the sex expression of hemp affects the quality of the fiber, where, traditionally, a good fiber quality is obtained with a male plant in dioecious varieties. Nowadays, the vegetal development of hemp is focused on improving monoecious varieties, due to the fact these varieties are better for fiber and seed production [27]. Even though there is significant similarities between fiber characteristics among the two hemp genotypes, there are also differences in the midst of the same genotype due to environmental conditions and extraction methods, which will be discussed further in this review, and also in agriculture management, like it was discussed before, and in the harvesting process.

The harvesting process is reported as one of the main factors contributing to the profitable development of the hemp supply chain, due to a lack of developed machines suitable for the morphological characteristics of hemp, leading to a high level of specialization and thus increasing the overall cost of harvesting [28]. Therefore, one of the main problems with hemp harvesting for fiber applications is the clogging of improper machines due to tough fibers [29]. Hence, some modifications or device couplings to the current machines used for harvesting hemp stems are necessary [28]. Therefore, the first step in harvesting involves cutting the stems and air-drying them in the field, and this can be achieved using a sickle bar mower to lay the hemp [17,29]. During the retting process, or the drying process in the field, which will be discussed further in greater detail, the stems need to be turned using a wheel rake before baling either in big round bales or big square ones for transportation. Now, there are developments in this area to join cutting and bailing [17,28–30]. In harvesting hemp for seeds, combine harvesters are normally used for corn production. There are some constrains to this process, one being that is hard to process plants higher than 2.5 m with this equipment, and, additionally, excessive wrapping can occur, leading to a clogging up of the machine. Also, hemp grain must be harvested with more moisture for less shattering to occur than crops like corn. Therefore, alternatives have been suggested to help overcome these problems, such as the introduction of other equipment like draper heads [17,28]. An important breakthrough in harvesting industrial hemp is to harvest the hemp seeds and leaves and simultaneously cut the stems for fiber extraction. This type of machine involves a double-cut combine, where the top part is a conventional combine made to harvest grain, like corn or barley, and the bottom part is a sickle bar mower, just as mentioned before, which cuts the stalks, leaving them on the field [30]. Nowadays, there has been an improvement of the machinery in this direction, allowing farmers to have two streams of income coming from the same plant, making hemp production more attractive for farmers. However, these types of machines are expensive and are not available for ordinary farmers who want to initiate hemp cultivation [28]. Therefore, additional developments are needed in this field of expertise to lower harvesting costs, because a double harvest is an attractive way method hemp cultivation in economic terms, as well as from an environmental point of view.

Growing industrial hemp is a key factor in obtaining a good fiber quality, thus the importance of including this sector in this review, but there are hurdles in the development due to legal restrains that have caused a hold on the further expansion of adequate machines and procedures to increase yield production and also the crop value. Despite all these challenges and problems, hemp could become, in the close future, an important crop in agriculture production [31].

### 2.3. Retting Process

The retting process, just as referred to in the previous section, is an important and necessary step to obtain fibers from stems or leaves, where the focus of this treatment is to remove the pectin holding the fibers together [32]. Through the removal of pectin, the goal is fiber extraction while maintaining the fiber morphology and mechanical potential [33]. In the beginning, this happens through the gradual weakening of the interactions among the fiber bundles and the surrounding material. Thus, the withdrawing of any non-cellulosic compounds is necessary to obtain a cellulosic-rich fiber [34–37]. Generally, retting involves the colonization of microorganisms on the plant, leading to a partial degradation of its constituents. Moreover, this process is affected by various factors, such as the plant's developmental stage and environmental conditions, which play a role in determining the quality and yield of the fibers [9]. Furthermore, an insufficient retting, or under-retting, leads to an incomplete degradation of the compound's matrix, minimizing the process's efficiency and fiber quality, whereas excessive retting, or over-retting, causes a higher removal of non-cellulosic components, leading to a decrease in the fiber's strength and losing possible applications in some sectors of the textile industry [9].

Nowadays, is possible to divide retting into four different categories: physical, semi-physical, chemical, and biological. In most cases, the retting process combines two or more of these categories to achieve a higher fiber quality [9,38]. The choice of retting method depends on factors like fiber quality, processing time, and environmental impact. In recent years, researchers have been exploring new methods and refining existing ones to improve efficiency, reduce the environmental impact, and enhance fiber properties [39]. These advancements may include the development of new enzymes, microbial cultures, or modified chemical processes to optimize the retting process [40]. It is essential to stay updated on the latest research in this field to be aware of any new and improved retting methods that may emerge over time. Researchers and industries are continually working towards more sustainable and efficient processes in the production of plant fibers.

Dew retting is the most used treatment, due to the fact that it has a lower cost and is simple to apply [38]. This type of treatment consists after harvest to lay down the stems in piles for 2 to 3 weeks [35,41]. Throughout this period, the action of the weather conditions, dew, precipitation, and sunlight, combined with the colonization of microorganisms, leads to a gradual degradation of the nonessential components [35,37,38]. Consequently, microorganisms develop on the plant's surface, degrading the superficial tissue by the release of specific enzymes, mainly pectinase, but also hemicellulases and cellulases [32,35,38,41]. In more detail, organisms of the Fungi kingdom and bacteria are present in this treatment, and it is reported that degradation happens in succession [35]. Therefore, the first microorganisms to colonize the plant are different types of fungi, capable of breaking through the cuticular layer with cutinases, as well as with hyphae through damaged areas. Shortly after, colonization by different species of bacteria occurs, taking advantage of damaged points created by fungi, and together metabolize the parenchyma cells between fiber bundles [35]. In later stages, these bacteria will start to damage the fiber's cellulose by releasing cellulosic enzymes, which are necessary to remove the plant from the field at the exact moment so over-retting does not occur [32,35]. Therefore, dew retting, as said before, shows its advantages as low cost and simple, and also, in soil enrichment, foul smells are avoided, with a lower environmental impact due to lower energy consumption and water utilization [35,41]. However, it shows some disadvantages, such as making the fields unworkable for close to a month and irreproducibility between harvests, mostly due to weather changes, which leads to changes in the fiber quality [33]. All in all, dew retting is a crucial step in obtaining fibers from stems; nonetheless, it is an empiric method, depending totally on weather conditions, leading to inconsistencies in the fiber quality. On the other hand, a controlled process of this kind, in a rigorous manner, could lead to superior and equal qualities from different harvests [38].

Alongside dew retting, water retting is a common procedure used as a retting method [41]. This method consists of submerging plants in artificial bodies of water or natural reservoirs

at a temperature between 15 °C and 30 °C for 5 to 7 days for hemp, because this period is dependent on the type of plant [34,35,38]. Water retting is one of the most ancient practices in the world for retting proposes, its utilization reported prior to 500 BC in the Himalayas region [33]. In further detail, it is a relatively moderate treatment, as it does not need any catalyst and does not cause significant corrosion problems, consisting of bacterial action and some fungi, present on the plant surface, for the degradation of soft tissues, mainly pectin [34,35,41,42]. At the bacterial level, the process is initiated by aerobic bacterium from the genera *Bacillus* and *Paenibacillus*, and when air runs out, the subsequent retting is made by anaerobic bacteria [35,41]. as Similar to dew retting, water retting has its advantages and disadvantages. The advantages consist of consistent yields of elevated fiber quality, shorter treatment time, the influence of weather and geography are minimized, and variables such as temperature and pH can be maintained in artificial bodies of water [35,41]. While finer and stronger fibers are obtained through this treatment, there are some problems, such as strong smells; weather-dependent if done in natural locations; and higher costs associated with water, the drying of fibers, and the treatment of residual water [35,41]. Even though water retting has been used for a long period of time, it is still an extremely empiric treatment, dependent on the flora present in the fibers; thus, a higher control is necessary to confine the cellulose degradation [34]. Therefore, it is possible to use variations to this treatment, such as hydrothermal, which consists of using water in its liquid or gas state to treat the lignocellulosic material, where high water pressure penetrates the biomass, removing pectin, hemicellulose, and lignin [42].

Another commonly used retting process focuses on the use of chemicals for the removal of non-cellulosic components from the fibers, in which sodium hydroxide is the most used. This type of retting is called chemical retting, made in an alkaline medium at high temperatures, jointly with detergent and soap like glycerol, ethers, or other solvents [9,36]. This treatment can be made in a continuous or discontinuous manner, contributing to excessive organic matter in textile effluents. These wastewaters possess disinfectants, detergents, insecticide residues, pectins, fats, oils, and ashes, among other chemical compounds, making the pH extremely high and resulting in a significant environmental impact [43]. At the same time, a high cost, the use of large amounts of chemicals, high temperature and pressure, and some significant damage to the fibers are associated with this type of retting [36,44]. However, chemical retting shows some advantages, like a short processing period and lower cost compared to other retting processes, such as biological ones, and also showing consistent results [32,41]. Additionally, in the last few years, different studies have focused on making these methods more sustainable and the conditions less aggressive by changing the chemicals applied but also lowering the temperature and pH value, making the process more sustainable, with lower damage to the fibers [36,44]. Alternatively, there is the possibility of using biochemical processes to overcome environmental issues. This alternative consists of the combination of chemicals with enzymes produced by modified microbial strains, like the use of enzyme alkali pectinase with additives such as TEMPO (2, 2, 6, 6-tetramethylpiperidine-1-oxyl radical) [36,44].

An emerging retting process is biological retting, which is the use of free enzymes or microorganisms as a whole to degrade nonessential components. This method is also known as enzymatic retting and has a low environmental impact when compared to other methods [36,37]. This type of treatment can be applied directly on the plant just after harvest or material removed after a brief retting period through water or dew retting [35]. Enzymatic retting involves pectinolytic enzymes in the degradation of the fibers' pectin, which hydrolyzes the glycosidic bounds in the homogalacturonan structure in monomeric, dimeric, or olimeric fragments [44,45]. This enzyme activity is dependent on retting conditions, like temperature, pH, inhibitors, and catalysts, among others, which affect the enzyme's performance, consequently affecting the fiber's treatment [44]. Using enzymatic retting, it is possible to observe some advantages, such as a lower environmental impact, high specificity, and standardized results, due to regulated protocols [33,36,41]. Nonetheless, there are some disadvantages, like high requirements of microbial strains, low

production capability, and instable enzymatic activity [36]. Despite all these advantages and disadvantages, promising results, and commercial availability of enzymes, there is a lack of application at the industrial scale, due, mainly, to the enzymes' high cost [35,46]. Nowadays, there has been a development towards application on an industrial scale and the elaboration of protocols, mainly in plants like flax and hemp [35]. In further detail, there are global research programs involving universities around the world, for example, connecting universities from Heilongjiang Province with universities in Canada and the Ukraine, with the aim to develop new technologies and protocols for enzyme usage at an industrial scale in hemp fibers by introducing biotechnology strategies, thus obtaining a more sustainable fiber [36].

In addition to traditional retting methods, recent innovations have introduced new approaches that leverage computer analysis and metabolic studies of microbiota found in natural environments, such as water and dew retting. These methodologies employ metagenomics, which involves investigating the genetic material present in environmental samples using molecular tools like 16S rRNA gene amplification. By applying the techniques to processes like hemp or flax retting, researchers aim to identify superior bacterial strains with enhanced enzyme activity crucial for selective retting, such as pectate lyase, pectinase, hemicellulose and ligninase, while minimizing cellulase activity [47–49]. This focus on specific enzyme activities preserves the natural structure and properties of the fibers, resulting in higher-quality fibers suitable for high-end textile applications [50–52]. Metagenomics holds promise in addressing various challenges associated with different retting methods. It can help achieve more consistent results lacking in dew retting, reduce water pollution in water retting, lower the costs associated with enzyme use in enzymatic retting, and eliminate harsh chemical usage in chemical retting [39,50,53,54]. Moreover, metagenomics-based approaches offer scalability advantages, as selected colonies of microorganisms in artificial tanks can achieve superior fiber quality in shorter durations compared to traditional methods, harvest after harvest [50]. While metagenomics has shown promising results in bast plants like flax, jute, or kenaf, its application in hemp retting remains relatively underexplored. Nonetheless, initial studies indicate potential benefits, including reduced retting time, increased fiber recovery rates, and enhanced fiber tenacity compared to traditional methods like water retting [39,54,55]. Utilizing metagenomics in hemp retting could lead to a more consistent and streamlined process, yielding higher-quality fibers and offering an easier scalability compared to recent methods like enzymatic retting.

In summary, there are different retting methods of natural fibers, such as the ones referred to prior, which play an important role in obtaining fibers with enough quality to be used in the textile industry [10]. Still, there is progress to be made, especially in terms of consistent results requiring little to no variation between harvests, maintaining a higher fiber quality.

### 2.4. Fiber Extraction

Following the retting stage, a critical and common step in fiber-related applications is fiber extraction. Typically achieved through mechanical means, the primary goal is to separate the woody core and obtain individualized fibers. Successful fiber individualization requires the retting process to be stopped at the appropriate time, allowing the stems to dry and facilitating the contraction of fiber bundles for easy release [56]. The choice of fiber extraction method depends heavily on the desired fiber properties, in which length and fineness are the most critical. Long hemp fibers, between 50 and 70 cm long, can be obtained to be processed in flax machines and produce short fibers as a by-product [56,57].

Decortication is an extraction method that allows obtaining fibers in a more direct way and without the need for alignment of the stems to feed the machine or without the retting process [48]. This process tends to break the nucleus, allowing fiber separation and obtaining fibers with a low level of individualization and high level of impurities, making these fibers not suited for high-value applications such as apparel textiles but applicable

in nonwovens production [56,58]. The resulting fibers of the decortication process can still suffer a process of degumming to remove pectins, making them more suitable to be processed in a carding machine, with properties similar to cotton in terms of length and fineness [56]. Moreover, one of these machines' greatest advantages is the high process capability with the use of hammer mills, where a single machine can reach a productivity of several tons per hour [56–60].

There are alternative methods to obtain a high fiber quality, which are based on the production of flax fiber with better value [59]. In this process, there is the need for an efficient retting process so that the breaking and separation of fibers happen with greater ease. Additionally, there is a need to align the stems parallel to each other when feeding rolls into the machine [56]. The first step is similar to decortication, whereas some authors do not make a difference, where it involves breaking the hurds inside the stem along the length of the plant. The system is made of rolls that crush and break these hurds into small portions in a way that allows them to be removed in the next stage [56]. This next step is called scutching, which is based on the beating of blades on a disk across its length in a way that removes its impurities, as well as promoting the individualization process of fibers, making them thinner and softer. The hurds are rubbed and crushed by the rotating tambour and separated by gravity. This process demonstrates high efficiency, achieving a rate of 500 kg per meter of the working width [56,60]. After this process, there is a final step, the hackling process, where the fibers are combed with combs progressively finer to align the fibers and reduce their diameter without reducing their length. The combs are supported on two rotating belts parallel to each other with adjustable speeds [56,59]. Some authors demonstrated that, when using the process described previously with an optimized retting process, it is possible to obtain hemp fibers with a quality and mechanical properties comparable to flax fiber, for which these processes are developed [37].

La Roche, Formation AG, Canadian Greenfield, Hempflax, Hempterra, Tatham, Canna systems, and HPP are examples of decorticating equipment suppliers. Cretes, Depoortere, and Vanhawaert produce and commercialize industrial machines dedicated to obtaining long fibers.

After fiber extraction, the following steps are dependent on the final application, visible in Figure 2, where the use of hemp fibers for nonwovens is simpler and does not require a good-quality fiber, as referred to before [12]. For apparel and home textile applications, the scenario is different, where a greater number of steps are needed and the spinning process is dependent on the type of application and fiber properties [12,19]. Long fibers are usually considered the best quality of fibers when regarding flax/hemp, although specific spinning processes are required, namely wet or semi-wet spinning processes, allowing 100% hemp yarn production. Regarding the semi-wet spinning process, a pretreatment of roving is required. In both cases, there is also the possibility of making blends with flax fibers [12,19,56]. In addition, through means of the cottonization of hemp fibers, as shown in the next figure, is possible to use ring and rotor spinning systems, making feasible the blend of hemp fibers with other fiber types. However, normally, using these systems is not possible to obtain 100% hemp yarn [19]. Lastly, there is the possibility to use wool systems to produce blended hemp/wool yarns. To achieve such prowess, hemp fibers must be decorticated but also pass through a specific carding machine, which enables them to obtain wool-type hemp fibers [19,56]. Through this spinning system, it is possible to achieve yarns with 90% hemp fibers [56].

All things considered, fiber extraction is an important step but a complex one, due to the fact that it depends highly on the final application of the fibers. Different extraction methods highly affect fiber characteristics, so it is important to know what the desired mechanical properties/applications are to choose the best extraction process available.

*2.5. Properties of Hemp Fiber*

An important point after fiber obtention passes through determining its physical properties, due to the fact it will affect the possible use and technologies applied in the

following steps until the final desired product [61]. These same properties are determined by the structural fiber characteristics, such as length and tertiary structure, among others [9]. Furthermore, in natural fibers, the extraction location will affect the geometric dimensions, where fibers from fruits (e.g., coconuts) and seeds (e.g., cotton) are a few centimeters long, but fibers from stems (e.g., hemp) and leaves (e.g., bananas) can reach over a meter in length [1,8,62]. In addition, the chemical composition, maturity, geographical location of the plantation, soil micronutrients, and environmental conditions will affect the mechanical, physical, and chemical properties of the fibers [62–64].

Hemp fibers are considered bast fibers, just like flax or nettle, which are composed of bundles that run along the plant length, surrounding the hurds, or shives, which are the woody core of the stem. All of this is covered by a thin layer, the epidermis, which is also known as the bark (Figure 3) [12,19]. Hemp fibers are built from several layers, the first a primary wall developed during cell growth consisting of three stratums and a middle layer giving shape to the mechanical fiber properties. The second layer, or the middle one, resides in a long cellulose chain containing between 30 and 100 molecules making up microfibrils, giving the mechanical strength to the fibers. This cellulose is the most desired component, thus ensuring the fiber's strength but also flexibility. There is also the presence of hemicellulose, which binds cellulose and lignin together; the latter increases the cell wall stiffness and hinders its degradation. Finally, pectin occurs in the middle layer among all cell types holding everything together, thus the importance of its removal to free the cellulose fibers [19,32]. All these different chemical compounds make up the whole structure of the hemp fiber, and they represent distinct percentages of the fiber itself. The most common is cellulose, which consists of anywhere between 52% and 78%, followed by hemicellulose, having a content of 14% to 22%, and lignin and pectin follow up as the remaining significant molecules in the bast fiber, representing, respectively, between 3% and 10% and 0.6% and 10% [19,35,65]. To determine such percentages, different techniques are used, mainly focusing on destructive analyses in the case of cellulosic fibers. Therefore, methods like High-Performance Liquid Chromatography coupled with Mass Spectrometry are good techniques that permit analyzing complex mixtures with great sensitivity. Alternative procedure such as Multi-Shot Pyrolysis coupled with Gas Chromatography and Mass Spectrometry or Evolve Gas Analysis coupled with Mass Spectrometry are other promising techniques that can be used to determine the constitution of cellulosic fibers [66]. The values depicted and achieved with the different methods for hemp are closely related to the chemical composition shown for other bast fibers, like flax or jute, thus demonstrating a possible connection between the similar applications and their morphology [35]. Additionally, and through the chemical composition, hemp is characterized as a cellulosic fiber, which means that post-fiber extraction processes, like pretreatment, dyeing, post-treatment, and finishing, are like those used in other common cellulosic fibers, like cotton or flax [67]. Therefore, the same chemicals can be used, like caustic soda, oxygen peroxide, and the same pigments, in some stages, but bigger concentrations may be needed in some cases due to the higher present of compounds like lignin, which leads to the need for a stronger pretreatment [68,69].

Altogether, it will affect the mechanical properties, such as tensile strength, tenacity, and elongation, but also characteristics like density, length, and diameter [8,9,61]. The fiber length is considered a crucial point, because the fibers need to be spun, and its length will affect the type of spinning that can be used. In the textile industry, a length over 15 mm is considered advantageous [9,19]. In hemp fibers, long fibers are preferred for numerous reasons, like easier to process, less pilosity, and a more consistent production of yarns with a higher tenacity [64,69]. Apart from its length, the diameter is equally important to determine its mechanical properties, mainly the tensile strength [8]. Natural bast fibers like hemp, flax, and nettle show characteristics that reveal the possibility of being an excellent source of fibers to be used in apparel and home textiles [62]. Additionally, for natural fibers, density is an essential factor that will affect the materials' mass. Generally, fibers originating from natural sources possess relatively low densities and have better elastic

properties when compared to carbon and glass fibers [8,70]. Elongation is another crucial parameter, and it is defined as the percentual increase of the fiber's length because it makes the fiber resistance to external forces [70]. Tensile strength is one of the most important aspects and can be used to compare performances between different fibers [61]. This fiber mechanical resistance is, consequently, associated with the yarn quality, where weaker fibers can lead to breaks during the spinning process [61,70]. Having these characteristics in mind, the next table (Table 1) shows it is possible to understand the variety of the different fibers, having, in some cases, big intervals of values associated with the fiber diversity due to several factors that do not exist in manmade fibers. However, this fiber has other advantages, which leads to an interest in the textile market for such fibers.

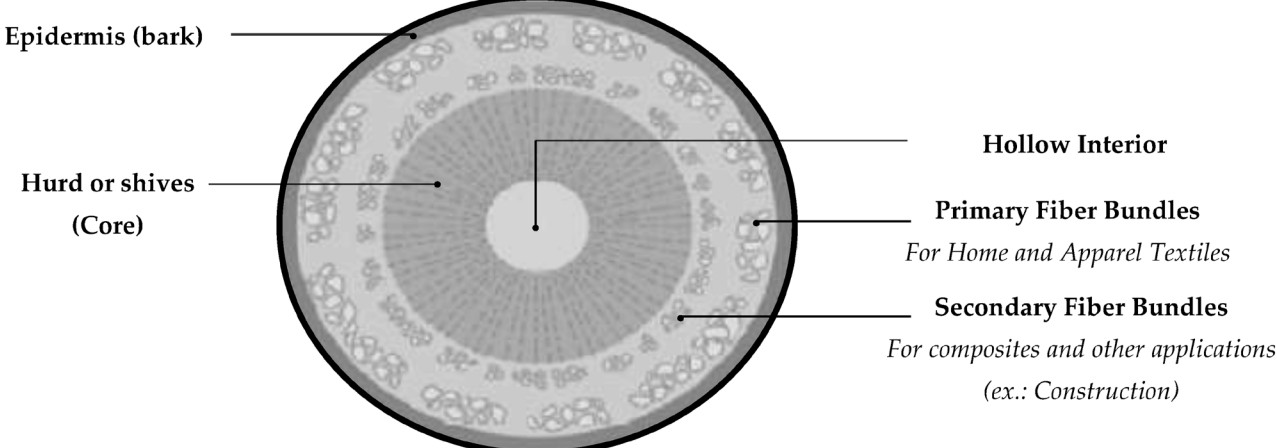

**Figure 3.** Morphology of bast fibers—more precisely, hemp stems. Based on [12].

**Table 1.** Physical properties of cotton, hemp, flax, and nettle fibers.

| Properties | Cotton | Hemp | Flax | Nettle | References |
|---|---|---|---|---|---|
| Fiber Length (mm) | 10–60 | 2–500 | 5–1000 | 16–171 | [9,32,41,59,62–74] |
| Diameter (μm) | 12–45 | 16–50 | 5–38 | 19–72 | [62,73–78] |
| Density (g/cm$^3$) | 1.5–1.6 | 1.4–1.8 | 1.5 | 0.7–1.5 | [41,73,74,78–83] |
| Linear Mass (dTex) | 1–3.2 | 3.3–30 | 6–40 | 18–27 | [19,75,76,83–88] |
| Tensile Strength (Mpa) | 287–800 | 550–1110 | 88–1500 | 300–1500 | [9,19,65,73,79,89] |
| Tenacity (cN/dTex) | 3.6–8.2 | 2.5–8 | 2.3–11 | 2.4–6.0 | [19,73,74,79,89] |
| Elongation (%) | 3–10 | 1.6–3.3 | 1.3–3.3 | 1.6–2.9 | [41,61,63,73,77,89] |

In the previous table, we can see the superior tensile strength of hemp and flax when compared to cotton, and although nettle has a similar range, its lower tenacity when compared to these fibers lowers the mechanical proprieties of said fiber. Additionally, it is possible to understand the higher range of values of the bast fibers when compared to cotton. This could be explained by the scope of the length, which can lead to a higher variability of values. While comparing hemp fibers to synthetic ones, such as polyester, it is possible to understand the superior proprieties of polyester, with mechanical proprieties of tenacity 10× times higher and elongation at around 20% while having a linear mass lower than 1 dTex [90]. However, this fiber has a production process with a very high environmental impact, thus increasing the interest in natural fibers like hemp [2,5,6,8].

*2.6. Hemp's Lyfe Cicle Assessment*

Hemp is considered by many as an alternative fiber due, mainly, to its supposedly positive impact on the environment and characteristics, just as referred to in the previous

section, although there are still some concerns about this affirmation and how greener hemp is in comparison to crops like flax, which is used many times as a comparison step to hemp. Therefore, this review investigated recent and emerging studies on Life Cycle Assessments (LCAs), which are particularly important to the brands or anyone claiming hemp as a positive plant for the environment, such as referred to previously concerning the carbon sequestration capability of hemp [12].

Firstly, carbon footprints often associated with hemp and LCAs cannot be compared easily, because both sets depend highly on the systems' boundaries, functional unit, and data, among other important parameters. Furthermore, one important problem with studies done regarding LCA tools for natural fibers is the difficulty in obtaining available data for specific bast fibers like flax and hemp. Additionally, assumptions regarding agricultural yields and practices have a considerable impact on the overall LCA results [91].

Therefore, most hemp-related LCA studies have focused on the production of nonwovens, then the apparel and home textile sector. This could lead to some miss conclusions about the hemp chain value, because, as seen in Figure 2, it is possible to observe that the latter is more complex and versatile than the nonwovens sector, which could highly affect the overall environmental impact of hemp, although there have been some studies that focused on the agricultural growth and impact of hemp. One study comparing a diverse sample of bast fibers (hemp, flax, jute, and kenaf) showed that there are not any significant differences between the agricultural impact of the four plants. Additionally, it showed that 1 ton of natural fibers is produced per 400 kg of $CO_{2\text{-}eq}$, but it also demonstrated that fertilizers have a huge impact, and changing for organic fertilization could further reduce the $CO_{2\text{-}eq}$ emitted by around 40 kg per ton [91]. Another solution for lowering the impact of hemp production is the simultaneous harvest of seeds and fibers, just as proposed in this article, which shows this idea is an alternative solution for the hemp textile sector [91].

Regarding the production of nonwovens, like hemp composites, some LCAs show savings of greenhouse gas emissions from 10% to 50% when compared to counterparts made from fossil fuels, and when taking into account carbon storage, these values could reach between 30 and 70% [91–95], although there are still some gaps to fill in emissions, especially for the different retting processes discussed earlier [86]. Furthermore, the substitution of manmade materials for hemp fibers comprises 66% of the total volume reduced and 45% of the energy needed for the whole process. However, water usage and emissions related to phosphates and nitrogen oxide are higher in these types of materials due to the use of fertilization, which grants a higher importance to the change of fertilization [92,93]. Still, in this study, the volume of hemp was 66% but only accounted for 5.3% of the cumulative energy and composite emissions [91]. Furthermore, hemp has high carbon storage potential, where a value of 325 kg of carbon per ton of hemp can be stored during the product's useful life. This can be translated into a reduction of 3 tons of $CO_2$ emissions when using hemp and 1.19 million $m^3$ of crude oil composition when applying hemp comprising as high as 50% of the production [92].

On the other hand, to produce apparel and home textiles, hemp currently does not have an overall lower impact than regular cotton textiles. This comes mainly from the use of old and outdated machinery and technology, which is detrimental to the environmental score of hemp textiles. Taking, for example, the addition of 55% of hemp fiber compared to cotton using the same technology, there is a reduction in some aspects of the environmental impact. Although there is a bigger overall damage to the environment, this does not support many claims made about hemp textiles; still, this information should be taken with caution, since there are many gaps in the data for this textile sector regarding hemp [96]. However, hemp has the potential to improve the environmental performance of the textile industry due to the fact that, agricultural speaking, and when compared to cotton, its cultivation has a reduction between 50% and 90% in all categories; even with nonorganic fertilization, hemp still produces better results [94]. The only hurdle for hemp textiles is the degumming phase, where it increases significantly when compared to cotton, drastically lowering its environmental performance. Thus, the development of better and greener

technologies for hemp degumming is needed, like enzymatic degumming and the use of alternative greener energy sources [37,93].

All in all, current hemp cultivation exhibits a comparable environmental impact to other bast fibers like flax. The improvement in yields, not only in agriculture but also in every processing step, just as referred to before, would increase hemp's environmental benefits [96]. Both hemp and flax are characterized by low-input and low-impact crops. However, while flax has undergone significant development over the past two decades with maximized yields, hemp has seen limited progress in recent decades, placing it at a similar stage in terms of agricultural impact. The belief in hemp's potential for development persists, and it is considered a very sustainable crop at present, though not yet an environmental miracle for the entire value chain in the context of apparel and home textiles. The prospect of hemp becoming an environmental wonder in the coming years hinges on its continued development, as observed in recent years [95–98].

## 3. Conclusions

In summary, this review underscores hemp's resurgence, particularly in response to the textile industry's increasing reliance on synthetic fibers and its commitment to achieving carbon neutrality. While hemp is gaining momentum as a versatile and sustainable resource, it remains a niche market facing challenges, primarily due to governmental restrictions on cultivation and harvesting. The limited knowledge in hemp cultivation and processing, especially when compared to more established fibers like flax, poses obstacles to its widespread development.

The versatility of hemp extends beyond textiles, with potential applications in food, cattle raising, cosmetics, and construction. Within the textile industry, hemp fibers emerge as a promising solution, offering rapid growth and favorable physical properties to address existing gaps.

Retting methods, specifically dew retting and water retting, are considered encouraging, with the latter requiring sustainable adaptations. Mechanical processes for fiber extraction are preferred, contingent on the intended application. The cottonization of hemp fibers presents an appealing alternative, despite its impact on the fiber characteristics.

The lack of comprehensive information on hemp's LCA in the textile industry, specifically in the apparel sector, is acknowledged. While some studies indicate positive environmental impacts in composites and nonwovens production compared to synthetic materials, further research is needed for a more comprehensive understanding.

Hemp has gained significant traction across various industries, particularly in apparel and technical textiles, owing to their unique properties and sustainability benefits. In the realm of fashion, hemp fibers are embraced for their breathability, which helps regulate temperature and ensures enduring comfort. Their exceptional durability and longevity make hemp clothing a preferred choice for consumers seeking both quality and a reduced environmental impact. Additionally, the inherent antibacterial properties of hemp fibers make them ideal for activewear and undergarments, promoting hygiene and odor control. Moreover, hemp's versatility allows it to be seamlessly blended with other materials, facilitating a broad spectrum of styles ranging from casual wear to formal attire. In the domain of technical textiles, hemp fibers are utilized in various applications such as automotive interiors, geotextiles for erosion control, and construction materials such as hempcrete. Their notable attributes, including strength, resistance to UV radiation, and biodegradability, make them suitable for a wide array of industrial uses, spanning from sports equipment to industrial filters. As research and development efforts continue to explore advanced processing techniques and innovations, the potential applications of hemp fibers in these industries are expected to expand further. This ongoing exploration reinforces hemp's position as a sustainable and adaptable material poised to play a crucial role in shaping the future of apparel and technical textiles.

In terms of environmental impact, hemp cultivation shows promise, surpassing cotton in positive contributions by around 70%. However, the slower development of hemp

cultivation compared to flax results in lower yields and environmental scores. Continued developments and data collection in this area could unveil the true environmental impact of hemp, offering insights into potential changes in the coming years.

In conclusion, despite the challenges, the evident sustainability and attractive characteristics of hemp create a growing probability of its increased cultivation and utilization in various industries, particularly in the realm of textile fibers.

## 4. Future Directions

As we anticipate the years ahead, hemp fibers development faces both challenges and promising opportunities for enhancement. Throughout this review, a critical issue that emerged is the widespread lack of differentiating industrial hemp from its medicinal counterpart. Historical misconceptions have resulted in lingering regulations and prejudices, hindering hemp cultivation and limiting the textile industry's access to valuable feedstock. To pave the way for industrial hemp's future, a crucial step involves educating farmers, government entities, and the public about the differences between hemp types and the myriad benefits it brings.

One of the most significant weaknesses in hemp production lies in the dearth of specialized modern machinery designed for comprehensive hemp processing, spanning from field to fiber to yarn. This deficiency, especially evident in hackling and spinning machines dedicated to producing high-quality long hemp fibers, can be traced back to years lost under restrictive regulations.

Agriculturally, hemp's low profitability for textile production necessitates innovative approaches to harvesting. The simultaneous harvesting of hemp stems and seeds could offer farmers a dual income stream. Additionally, advancements in agrotechnology and biotechnology are crucial for booting hemp production yields, ensuring a higher percentage of quality fibers with consistent physical properties.

A more exhaustive examination of the environmental impacts of hemp production, including greenhouse gases emissions, water usage, fertilization, and pesticides, is imperative. Consumer usage aspects, such as the washing requirements of hemp textiles, and end-of-life considerations, including the recyclability of hemp products, warrant thorough investigation. Although textile recycling has seen significant advancements in recent years through different methods, such as mechanical, chemical, and enzymatic recycling, hemp fiber recycling is still in its early stages. Additionally, since hemp fibers are often used in composite materials, efficiently separating them from the matrix material to enable the recycling of both components is a crucial objective for the near future. Some studies have begun to explore recycling hemp fibers in apparel textiles using conventional methods like those used for cellulose recycling, yielding promising results. In cases where the traditional recycling of hemp proves challenging, alternative approaches such as converting hemp waste into energy through processes like anaerobic digestion or combustion offer viable solutions. This alternative could help in extracting value from hemp biomass that might be otherwise difficult to recycle conventionally. Therefore, a future perspective for hemp textiles involves the ongoing development and study of hemp fibers' recyclability and the impact on its physical properties post-recycling.

To optimize the high-value utilization of hemp from field to fibers, it is essential to begin with optimized cultivation practices, including varietal selection and efficient harvesting techniques. Choose appropriate retting methods during processing and invest in modern and proper equipment for fiber extraction. Diversify product development to include textiles, composites, paper, and construction materials. Target markets valuing sustainable products, collaborate with researchers for innovation, and explore waste utilization possibilities. Stay informed about emerging technologies, adhere to certifications and standards, and engage in education and advocacy efforts to promote the benefits of hemp. By addressing each stage with a focus on quality, sustainability, and innovation, the full potential of hemp can be realized, contributing to the growth of a thriving hemp industry. Exploring the full potential of hemp has led to new trends in innovation, for example, the

chemical modification of the surface of hemp fibers. This technique focuses on changing the behavior of the hemp fibers by using different treatments, like the use of NaOH, EDTA, or ions, thereby imbuing them with new proprieties, such as reduced moisture sorption and minimized fiber heterogeneity. By modifying such proprieties, the potential applications of hemp can be greatly expanded, enabling its utilization in advanced technologies or more demanding sectors such as automotive composites or load-bearing construction.

In conclusion, a promising trajectory awaits hemp in the textile industry's future, contingent upon continuous advancements in all sectors. Bridging the gap with other fibers demands the development of new and sophisticated processes, compensating for the setbacks incurred under stringent regulations. Despite the challenges, the potential for a vibrant future for hemp in textiles remains within reach.

**Author Contributions:** Conceptualization, J.M.; methodology, J.M., C.G. and L.R.; writing—original draft preparation, J.M.; writing—review and editing, C.G., T.L.S. and L.R.; supervision, L.R. and C.J.S.; and project administration and funding acquisition, C.J.S. All authors have read and agreed to the published version of the manuscript.

**Funding:** The authors acknowledge the financial support from integrated project be@t–Textile Bioeconomy (TC-C12-i01, Sustainable Bioeconomy No. 02/C12- i01.01/2022), promoted by the Recovery and Resilience Plan (RRP), Next Generation EU, for the period 2021–2026.

**Institutional Review Board Statement:** Not applicable.

**Data Availability Statement:** Not applicable.

**Conflicts of Interest:** The authors declare no conflicts of interest.

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
