# Peer review of "Hemp: From Field to Fiber—A Review"

_textiles, doi:10.3390/textiles4020011_

Round 1

Reviewer 1 Report (New Reviewer)

Comments and Suggestions for Authors

This article makes a comprehensive review from hemp cultivation, production to products, which is informative and pleasing to readers. The following suggestions would be helpful for further improvement.

1. When comes to hemp, the first product run in mind is the textiles. Thus, it is recommended to include more information or examples on hemp related textile production process such as pretreatment, dyeing, posttreatment and finishing.

2. Pls include the applications of hemp on advanced technology, which expands the applicable area of hemp.

3. Pls mention the cases on the modification of hemp.

Author Response

Dear Reviewer

Thank you for the comments made about our article, but also for the suggestions presented!

We ensured to incorporate each suggestion provided, particularly focusing on points two and three, which are pivotal for the future trajectory of hemp. Additionally, we have taken note of the suggestion regarding the modification of hemp, and we will certainly consider this aspect in our further research and discussions.

We have reformulated and completed the manuscript in both Section 2.5 and the "Future Directions" section.

Thank you for your valuable input, and we appreciate your attention to detail in improving our work.

Kind regards,

The authors

Reviewer 2 Report (New Reviewer)

Comments and Suggestions for Authors

The submitted manuscript brings various aspects of HEMP-derived fibers. The authors cover the commercial side of hemp cultivation, fiber extraction, hemp fire properties, and life cycle assessment. 

The author must add a few things to the manuscript to improve its readability and broadness of the review.

(1) Different stages of hemp processing require some use of chemicals (which are essential for the process); therefore, authors should provide more insights into such processes.

(2) Chemical composition of Hemp Fiber: What techniques are used to calculate the chemical composition of hemp fiber, as this is essential for various application-oriented goals? The author could summarise spectroscopic methods commonly used to calculate the fiber's chemical composition.

(3)  Fiber post-processing characterization and comparison with other natural or synthetic fibers.

(4) Provide more structural information on hemp stems. For example, bark-based fiber is composed of lignocellulosic content (Lignin, hemicellulose, and cellulose). Similar information must be provided for the hemp stems.

Author Response

Dear Reviewer

Thank you for the revision made on our paper, and for the comments made, but also for the suggestion. Thank you for emphasizing the importance of integrating points three and four, which provide valuable insights into understanding the behaviour of hemp in relation to other fibers and identifying any chemical differences between similar fibers. We have taken this into consideration and ensured to incorporate these aspects into our revisions. While we acknowledge the significance of all the points raised, our primary focus was on enhancing sections 2.5 and "Future Directions" to comprehensively address the suggested improvements comprehensively.

We appreciate your guidance in refining our manuscript and are committed to further enhancing its quality.

Kind Regards,

The authors

Round 2

Reviewer 1 Report (New Reviewer)

Comments and Suggestions for Authors

Authors have made changes on the MS which meets the publication criteria.

Reviewer 2 Report (New Reviewer)

Comments and Suggestions for Authors

Authors provide enough information, and included in the manuscript.

This manuscript is a resubmission of an earlier submission. The following is a list of the peer review reports and author responses from that submission.

Round 1

Reviewer 1 Report

Comments and Suggestions for Authors

The manuscript is very interesting, but according to the topic, it is not suitable for Textile journal. The main part of the manuscript is about growing hemp. Very little is given about the properties of materials and fibers, only textbook level information is given. Therefore, I think that the manuscriptis not suitable for this journal.

Reviewer 2 Report

Comments and Suggestions for Authors

In this paper, hemp from field to fibers, it mainly talks hemp processing stage including cultivation, harvesting, retting, fiber extraction. Overall structure of this paper is ok.

Other comments are:

The sentence “-75% of the stalk is hurds, having this many applications”, “this”→“these”?

For retting, four categories are listed: physical, semi-physical, chemical, and biological. Which method are used to produce harvest fibers with excellent properties? At present, is there new method for retting?

Figure 1 and Figure 2 can be merged into one Figure.

How to recycle hemp in the future is still a problem.

Hemp from field to fibers, how to make the high value utilization of hemp?

Reviewer 3 Report

Comments and Suggestions for Authors

In my opinion the article is of low importance. Majority of content is rather known. Significant part of the article concerns the hemp harvesting. It is out of scope of the Journal. It is rather agriculture. The part concerning the LCA is interesting and worthy of publication. However, there is lack of information about application of hemp fibres in different areas: apparel and technical textiles. It would be more interesting for Textiles' readers.

In my opinion, in current state the article should not be published in the Textiles journal.